# Wastewater-Based Epidemiology for Viral Surveillance from an Endemic Perspective: Evidence and Challenges

**DOI:** 10.3390/v16030482

**Published:** 2024-03-20

**Authors:** Marco Verani, Alessandra Pagani, Ileana Federigi, Giulia Lauretani, Nebiyu Tariku Atomsa, Virginia Rossi, Luca Viviani, Annalaura Carducci

**Affiliations:** Laboratory of Hygiene and Environmental Virology, Department of Biology, University of Pisa, Via S. Zeno 35/39, 56123 Pisa, Italy; marco.verani@unipi.it (M.V.); alessandra.pagani@phd.unipi.it (A.P.); g.lauretani@studenti.unipi.it (G.L.); nebiyu.atomsa@phd.unipi.it (N.T.A.); v.rossi33@studenti.unipi.it (V.R.); luca.viviani97@gmail.com (L.V.); annalaura.carducci@unipi.it (A.C.)

**Keywords:** wastewater-based epidemiology, wastewater surveillance, human adenovirus, enterovirus, norovirus, SARS-CoV-2, endemic environmental monitoring

## Abstract

Wastewater-based epidemiology (WBE) is currently used to monitor not only the spread of the viral SARS-CoV-2 pandemic but also that of other viruses in endemic conditions, particularly in the absence of syndromic surveillance. The continuous monitoring of sewage requires high expenditure and significant time investments, highlighting the need for standardized methods and structured monitoring strategies. In this context, we conducted weekly wastewater monitoring in northwestern Tuscany (Italy) and targeted human adenovirus (HAdV), norovirus genogroup II (NoVggII), enterovirus (EV), and SARS-CoV-2. Samples were collected at the entrances of treatment plants and concentrated using PEG/NaCl precipitation, and viral nucleic acids were extracted and detected through real-time reverse transcription qPCR. NoVggII was the most identified target (84.4%), followed by HAdV, SARS-CoV-2, and EV. Only HAdV and EV exhibited seasonal peaks in spring and summer. Compared with data that were previously collected in the same study area (from February 2021 to September 2021), the results for SARS-CoV-2 revealed a shift from an epidemic to an endemic pattern, at least in the region under investigation, which was likely due to viral mutations that led to the spreading of new variants with increased resistance to summer environmental conditions. In conclusion, using standardized methods and an efficient monitoring strategy, WBE proves valuable for viral surveillance in pandemic and epidemic scenarios, enabling the identification of temporal–local distribution patterns that are useful for making informed public health decisions.

## 1. Introduction

The outbreak of the SARS-CoV-2 pandemic in 2020 has renewed interest in wastewater-based epidemiology, which serves as both an early warning tool and a long-lasting surveillance method for a pathogen’s spread within a population. It was first suggested in the middle of the last century thanks to the research of Joseph Melnick, who focused on research on the poliovirus, the etiologic agent of poliomyelitis, in the wastewaters of the cities of Chicago and New York to evaluate the relationship between the environmental detection of the virus and cases of acute flaccid paralysis [1]. Environmental monitoring of poliovirus is still applied to evaluate the effectiveness of World Health Organization (WHO) strategies adopted for its global disease eradication program [2,3] and to evidence its possible resurgences by monitoring the efficacy of prevention measures and pointing out their failures. Sewage surveillance in 2022 highlighted the circulation of vaccine-derived poliovirus (VDPV) in some countries, including Great Britain [4], Israel [5], and the United States of America, and a genetic analysis revealed the association of the detected strain with a paralytic case in New York [6]. This information allowed the adoption of public health measures to prevent the spread of the virus in the most affected age groups; e.g., the UK government recommended a booster dose of inactivated polio vaccine in August 2022 for all children in London under 9 years of age [4]. During the COVID-19 pandemic, WBE surveillance was applied worldwide (e.g., in Finland, Australia, Bangladesh, India, Japan, and North America [7,8,9,10,11,12]) to improve systems for early warning and variant tracking in parallel with clinical epidemiological surveillance. WBE produced useful results in this context; therefore, there was an increasing interest in improving the detection methods, as well as in extending environmental surveillance to other viruses, especially endemic ones (e.g., enterovirus, norovirus, hepatitis A virus, hepatitis E virus, rotavirus, and adenovirus [13,14,15]). In fact, endemic viruses often lack clinical surveillance, and WBE could provide useful information about their circulation. In Italy, an example of the combined analysis of clinical and environmental data for endemic viruses dates back to 2006, when Carducci et al. [16] investigated the relations between some enteric viruses detected in feces specimens and their occurrence in environmental matrices, revealing the continuous environmental circulation of such viruses despite the fact that the clinical data did not demonstrate any epidemic peaks and despite the concordance of the adenovirus and rotavirus strains between clinical and water samples. In the United States, enterovirus monitoring in feces and sewage revealed similarities with clinical serotypes, thus allowing the forecasting of the clinical viral strains based on those found in the environment [17]. Then, after the COVID-19 pandemic, the application of WBE to enteric viruses (e.g., norovirus, enterovirus) prompted new interest in the early detection of outbreaks and the estimation of the prevalence of infections in endemic scenarios [18,19,20,21,22]. Nevertheless, the routine monitoring of sewage is expensive and time consuming, and its relevance for public health depends on the monitoring strategy (e.g., time, number, and area of sampling) and on the reliability of analytical methods, as widely described during last year’s COVID-19 pandemic (e.g., [23,24]). In fact, a review by Kallem et al. [23] underlined that, when applied to SARS-CoV-2 detection, WBE requires daily sampling and rapid analysis and reporting; moreover, to compare sewage viral data over time and among different communities, it is worth correcting such data according to wastewater flow and to the population size of the study area. On the other hand, the wide use of rapid molecular methods for viral RNA detection requires high efficiency and sensitivity and is strictly associated with the wastewater’s chemical composition, the presence of inhibitors, and a low concentration of the target. Also, Cianella et al. [24] demonstrated the wide range of sample concentration methods used in different studies, thus highlighting the importance of adopting a standardized protocol for the worldwide application of WBE. Such aspects also need to be carefully considered in relation to the epidemiological situation of the selected viral targets in both endemic and epidemic scenarios. This study aimed to collect information on viral circulation in an endemic scenario through one year of sewage monitoring for human adenovirus (HAdV), norovirus genogroup II (NoVggII), enterovirus (EV), and SARS-CoV-2.

## 2. Materials and Methods

### 2.1. Environmental Sampling Strategy

From October 2021 to September 2022, 197 weekly 24 h composite samples of raw wastewater were collected at the inlets of four urban wastewater treatment plants (WWTPs) in northwest Tuscany and were stored at 4 °C before analysis (within 48 h). The WWTPs chosen in this study were those involved in the surveillance network of the national project “Environmental Surveillance of SARS-CoV-2 by urban sewages in Italy” (SARI), operated according to an EU recommendation [25], as previously described [26,27]. Briefly, the studied WWTPs served populations of between 42,931 (WWTP1) and 110,871 (WWTP3), and there were some differences in the sewer network structures and population composition. WWTP1 had a separate sewerage system, with a small portion originating from a large hospital; WWTP2 served a small city with a large surrounding area, mostly through combined sewerages (89%); WWTP3 served a moderately sized industrial city with a separate sewerage network characterized by high rainwater infiltration; WWTP4 was situated in a highly touristic area where the population drastically increased during the summer season [27].

### 2.2. Sample Concentration

Raw sewage samples (45 mL) were pretreated at 56 °C for 30 min to inactivate infectious viral particles prior to being processed. The sample concentration process was performed according to the analytical protocol recommended by the SARI project, based on polyethylene glycol (PEG) and NaCl precipitation [28]. Briefly, after an initial centrifugation at 4500× *g* for 30 min, supernatant was mixed with PEG 8000 and NaCl. After shaking, the sample was centrifuged at 12,000× *g* for 2 h at 4 °C, then the pellet was resuspended directly in the nucleic acid extraction lysis buffer (2 mL) of the NucliSense EasyMag kit (BioMérieux, Marcy-l’Étoile, France).

### 2.3. Extraction and Purification of Viral Nucleic Acids

The extraction of viral nucleic acids was performed with the above-mentioned commercial kit, which allows the simultaneous extraction of DNA and RNA using magnetic silica beads, according to the manufacturer’s instructions [26]. Briefly, 2 mL of guanidine–thiocyanate-based lysis buffer was added to the pellet obtained after the concentration process. The solution was allowed to react at room temperature for 20 min, which facilitated the degradation of protein components of both cells and viral particles. Subsequently, 50 µL of magnetic silica beads was added to allow binding with free nucleic acids. The tubes were then washed three times while placed on a magnetic support. Finally, nucleic acids were released from the beads in an elution phase (100 μL). The final elution volume was purified using OneStep PCR Inhibitor Removal Kits (Zymo Research, Irvine, CA, USA) to remove PCR inhibitors. The extracts were then stored at −80 °C until quantitative determination of viral genomes as specified in Section 2.4.

### 2.4. Detection of Viral Nucleic Acids

Detection and quantification of viral targets were performed using previously published primer/probe sets (Table 1) and protocols, as described below. Real-time qPCR for HAdV was performed using Taq Man Universal Master Mix (Applied Biosystems, Foster City, CA, USA) in a total volume of 25 μL with 10 μL of DNA extract and primers and probe concentrations of 900 nM and 225 nM, respectively. The thermal protocol included the following conditions: Uracil DNA glycosylase activation at 50 °C for 2 min, AmpliTaq Gold DNA polymerase activation at 95 °C for 10 min, and then 45 cycles of amplification at 95 °C for 15 s and 60 °C for 1 min. The RNA virus genome of EV, NoVggII, and SARS-CoV-2 was screened by a one-step RT-qPCR using AgPath-ID™ One-Step RT-PCR Reagents (Life Technologies, Carlsbad, CA, USA) in a total volume of 25 μL with 5 μL of RNA extract. Primers and probe concentrations were 600 nM (primers) and 250 nM (probe) for EV, 1000 nM (primers) and 100 nM (probe) for NoVggII, 500 nM (forward primer), 900 nM (reverse primer), and 250 nM (probe) for SARS-CoV-2. The amplification conditions for both EV and NoVggII were reverse transcription for 30 min at 48 °C, reverse transcriptase (RT) inactivation at 95 °C for 10 min, and 45 cycles of amplification at 95 °C for 15 s and at 60 °C for 1 min. For SARS-CoV-2, thermal conditions were 30 min at 50 °C, 5 min at 95 °C, then 45 cycles of 15 s at 95 °C and 30 s at 60 °C. All reactions were performed in duplicate in 96-well optical plates using an ABI 7300 Sequence Detector System (Applied Biosystems, Foster City, CA, USA). For each target, viral titers (genomic copies—GC) were estimated using standard curves obtained by serial dilution of synthetic specific dsDNA (from 1.0 × 10^1^ GC/µL to 1.0 × 10^5^ GC/µL). The level at which their amplification was exponential was used as the threshold level that applied to the amplification plot of the samples. Samples with a threshold cycle (Ct) greater than 40 were considered negative. Samples negative for the presence of viral genome were considered equal to half of the limit of detection (LOD) obtained for each target virus by testing serial dilutions of standard dsDNA and by calculating the lowest genome concentration at which all replicates were positive [27]: 3 GCs/reaction for SARS-CoV-2, 2 GCs/reaction for HAdV, and 1 GC/reaction for EV and NoVggII.

### 2.5. Data Adjustment and Normalization 

The concentrations of each target virus were normalized to take into account the daily WWTP flow and the population served by the sewage system [27]. Therefore, the wastewater data were normalized using the following equation:$$Normalized\;viral\;load_{x}=\frac{Conc._{virus}\times \;Fd_{x}\times 10{\;}^{5}}{P_{x}}\;$$
where the following applies:-Normalized Viral Load (hereafter viral load) is expressed as GC/100,000 inhabitants/day, and *x* is the identification number of each WWTP (namely: 1, 2, 3, 4);-*Conc._virus_* is the concentration of virus detected (GC/L);-*Fd* is the daily wastewater flow rate of the WWTPs (L/day);-10^5^ is a constant used to relate the viral load to 100,000 inhabitants;-*P* is the number of inhabitants served by each WWTP.

### 2.6. Historical Data on SARS-CoV-2

Data on SARS-CoV-2 related to the same four WWTPs were extracted from a previous study [26] in order to cover three seasons in the 2021 period, namely, winter (20 data), spring (52 data), and summer (52 data). These data were compared with the monitoring data of the present study.

### 2.7. Statistical Analysis

For each virus, the chi-squared statistic was used to understand whether the presence of the virus (variable with two categories: presence or absence) was influenced by the sampling period, i.e., the season (variable with four categories: autumn, winter, spring, summer). For the present analysis, the seasons were divided as follows: autumn from 1 October to 31 December 2021, winter from 1 January to 31 March 2022, spring from 1 April to 30 June 2022, and summer from 1 July to 30 September 2022. The chi-squared statistic was also used to test the association between viral presence and sampling site, i.e., the type of WWTP (variable with four categories: WWTP1, WWTP2, WWTP3, WWTP4), separately for each virus. Regarding the viral loads, they were Log_10_-transformed prior to statistical analysis, and, similarly to as described previously for viral frequencies, a two-way ANOVA test was performed on the viral load to evaluate the influence of the sampling period (season) and of the sampling site (type of WWTP). The results were considered significant when the *p*-value was < 0.05, highly significant when the *p*-value was < 0.01, and extremely significant when the *p*-value was <0.001. All statistical analyses were performed using GraphPad Prism software (GraphPad, Boston, MA, USA, version 5.03, 10 December 2009).

## 3. Results

### 3.1. Descriptive Analysis of Virus Data

The occurrence of the target viruses and their viral loads, separately for each WWTP, are reported in Table 2 as pooled data and in Appendix A according to season.

Overall, 98.5% (194/197) of the wastewater samples were positive for at least one of the target viruses; NoVggII was the most prevalent (166/197, 84.3%), followed by HAdV (158/197, 80.2%), SARS-CoV-2 (100/197, 50.8%), and EV (87/197, 44.2%). For HAdV, the frequency of positive samples was statistically different either among seasons (chi-squared, *p*-value < 0.001) or among WWTPs (chi-squared, *p*-value < 0.01). In particular, the presence of HAdV was higher during spring and summer, with close to 100% positive samples in all WWTPs (Appendix A). Nevertheless, HAdV occurrence was lower for WWTP3 and WWTP4 during autumn and winter, which represent low tourist seasons for these areas. Similarly, EV-positive samples showed significant differences among seasons (chi-squared, *p*-value < 0.001) and among WWTPs (chi-squared, *p*-value < 0.001). The highest number of positive samples was observed during warmer periods (Appendix A), namely, 59.6% (31/52) in spring and 67.3% (35/52) in summer. Regarding the sampling location, WWTP2 showed a very low occurrence of EV, with no detection in autumn and winter and 15.4% (2/13) in both spring and summer, probably due to the structure of the sewer network (the combined sewage could be responsible for wastewater dilution; Section 2.1). For NoVggII (Appendix A), the presence of the virus was significantly influenced by the sampling location (chi-squared, *p*-value < 0.01), and, as reported for EV, WWTP2 was the least contaminated, with occurrence ranging from 61.5% (8/13) in summer to 80% (8/10) in winter. Finally, there were no statistically significant differences for SARS-CoV-2 either among seasons (chi-squared, *p*-value = 0.22) or among WWTPs (chi-squared, *p*-value = 0.17) (Appendix A).

Regarding viral load, HAdV showed the highest concentration with 9.4 ± 2.6 Log_10_ GC/100,000 inh/day, followed by NoVggII (8.5 ± 2.0 Log_10_ GC/100,000 inh/day), SARS-CoV-2 (6.5 ± 2.1 Log_10_ GC/100,000 inh/day), and EV (5.9 ± 2.2 Log_10_ GC/100,000 inh/day) (Table 2; Appendix A). As with viral occurrence, viral load also was influenced by sampling time (seasons) and sampling location (WWTPs), with different patterns depending on the type of virus (Figure 1). For HAdV, viral load was significantly higher during warm seasons, namely, spring (9.5 ± 2.7 Log_10_ GC/100,000 inh/day) and summer (10.8 ± 1.2 Log_10_ GC/100,000 inh/day) (two-way ANOVA, *p*-value < 0.01), also showing differences among WWTPs, with WWTP3 and WWTP4 being less contaminated (two-way ANOVA, *p*-value < 0.05). Similarly, EV load showed statistically significant differences according to season, with higher viral load in spring (6.7 ± 2.3 Log_10_ GC/100,000 inh/day) and summer (6.8 ± 2.1 Log_10_ GC/100,000 inh/day) (two-way ANOVA, *p*-value < 0.01), and according to WWTP (two-way ANOVA, *p*-value < 0.01), with lower contamination in WWTP2, as observed for EV occurrence. Regarding NoVggII, statistically significant differences were observed only according to WWTP (two-way ANOVA, *p*-value < 0.01), with WWTP2 being less contaminated, as reported for EV load. No seasonal pattern was observed. Regarding SARS-CoV-2, no significant differences among seasons or WWTPs were observed.

### 3.2. SARS-CoV-2 Annual Trend from 2021 to 2022

The availability of historical data for SARS-CoV-2 related to the previous year (February 2021–September 2021; Section 2.6) allowed a comparison with the 2022 surveillance period. This comparison revealed seasonal variability in SARS-CoV-2 load between 2021 and 2022. In 2021, SARS-CoV-2 load gradually decreased from winter (6.4 ± 2.3 Log_10_ GC/100,000 inh/day) to spring (5.7 ± 2.1 Log_10_ GC/100,000 inh/day) and summer (5.3 ± 0.9 Log_10_ GC/100,000 inh/day). Interestingly, in the subsequent 2022 period, such a seasonal pattern was not observed, with consistent viral loads of 6.8 ± 2.2 Log_10_ GC/100,000 inh/day, 6.7 ± 2.1 Log_10_ GC/100,000 inh/day, and 6.0 ± 2.1 Log_10_ GC/100,000 inh/day during winter, spring, and summer, respectively (Figure 2).

## 4. Discussion

Viruses transmitted by the fecal–oral route, such as enteric viruses, offer a unique opportunity for tracking emerging pathogens and investigating the epidemiology of infectious diseases within the community by studying viral shedding in wastewater. This could be used to study a wide spectrum of viral pathogens shed via the fecal–oral route, not just enteric viruses. Using such a method for epidemiologic surveillance would improve our understanding of virus circulation in different scenarios [32]. In our work, HAdV was detected during all 12 months. A statistically significant variation between seasons was demonstrated using either qualitative analysis based on the frequency of positive samples or quantitative analysis based on viral load, with peaks for these two parameters occurring in spring and summer. This finding is consistent with other work examining enteric viruses in stool samples [33], and suggests a higher level of HAdV DNA in summer, indicating a possible seasonal pattern. However, the literature also presents conflicting evidence, such as an insignificant seasonal distribution [34] or even a greater presence of HAdV in the winter months [35,36]. Our study was conducted in a part of Tuscany that is heavily influenced by summer tourism. This may have increased viral circulation due to a higher flow of people considering that approximately 90% of the human population is positive for at least one serotype of adenovirus [37]. The wastewater-based epidemiology of HAdV is crucial because, although the virus is the second most significant viral pathogen causing infant gastroenteritis and has been associated with outbreaks in various settings [37], there is a lack of consistent syndromic surveillance. Furthermore, in addition to the predominant presence of species associated with gastrointestinal symptoms (HAdV-F), Adenovirus C, responsible for respiratory infections, has also been observed in wastewater [38]. As reported in an epidemiological study of wastewater treatment plants located in different Italian regions, the analysis of 141 raw sewage samples showed a viral positivity of 60%, and the use of next-generation sequencing allowed the identification of up to 19 HAdV types, not only 40 and 41 (species F), but also viruses belonging to species A, B, C, D, and E, confirming the widespread presence of HAdV in raw sewage and, therefore, its circulation in the population with or without respiratory and gastrointestinal symptoms [39]. Similar results were obtained in Venezuela, in 2021–2022, by Zamora-Figueroa et al. [40], who performed a one-year surveillance study on 91 wastewater samples from urban areas of Caracas, revealing a positivity rate for HAdV of 52.7% with the presence of F and non-F species, equally distributed. In Egypt, the isolation and genotyping of HAdV in sewage and clinical samples from 2016 to 2020 revealed that no enteric species represented 5–6% of positive samples, thus suggesting their involvement not only in respiratory infections, but also in diarrheal diseases [41]. In China, Lee et al. [42] thoroughly investigated sewage samples collected from 2021 to March 2022, showing the presence of 14 HAdV C species, further confirming the role of sewage in the environmental spread of non-enteric HAdV with possible occurrence of recombination events and new variants in such matrices. Also in our study, the sequencing analysis of the positive samples revealed the presence of enteric and non-enteric viral species. Moreover, we found a prevalence of respiratory types in winter subsequently replaced by gastrointestinal types in summer [43]. This highlighted the importance of studying the prevalence of HAdV in aquatic environments to effectively monitor and prevent their transmission.

EV also showed a statistically significant difference in distribution across seasons and WWTPs, with the highest number of positive samples and higher concentration of viral RNA detected in the warm months (spring and summer) and in three out of four WWTPs. These results are consistent with previous studies, indicating a peak of EV in summer [44] or during summer and early autumn [45]. In temperate climates, the number of infections is highest during the summer and early autumn months as EV is a resilient organism that can withstand significant temperature fluctuations [46]. However, EV is also not consistently monitored through clinical surveillance, making it difficult to determine the exact number of infections at any given time. Investigating the presence of EV in wastewater and implementing a study with sequencing techniques could significantly improve the understanding of enterovirus infections. 

NoVggII and SARS-CoV-2 showed no statistically significant differences between months, neither in the frequency of positive samples nor in the viral load. Regarding NoVggII, our results are in disagreement with some of the literature, showing peaks in February and March [47], while SARS-CoV-2 is still under study. However, in the case of SARS-CoV-2, current evidence suggests increased mortality and infectivity at low temperatures, indicating a prevalence during the colder months. As a result, the circulation of SARS-CoV-2 is influenced by the seasonal cycle, as has been reported also for other respiratory viruses [48]. In fact, sunlight and high environmental temperature are responsible for viral inactivation; e.g., Sharun and collaborators [49] reported that solar radiation could have a significant impact on the rate of decay and viral inactivation. This aspect is supported by our data from the 2021 period, where a slight but consistent decrease in the viral titer was in wastewater from winter to summer. However, the seasonal trend tended to disappear in 2022, suggesting the emergence of new variants potentially more resistant to environmental conditions (e.g., high temperatures, UV rays). Such an observation is supported by the work of Gibson and colleagues [50], who observed that a 17% higher dose of UV254 is required for the disinfection of Delta and Omicron variants compared to the ancestral strain of SARS-CoV-2. In fact, in our study, the SARS-CoV-2 concentration did not vary much across seasons during 2022, showing a constant trend more similar to endemic situations, thus suggesting a possible transition of SARS-CoV-2 from an epidemic to an endemic situation, at least in the region under study. Furthermore, the endemic trend may also be facilitated by the increased transmissibility of the new variants. For example, the Omicron variant was found to be less virulent but more transmissible than the previous ones, thus promoting a higher viral circulation and, consequently, a higher number of infections. This is easily understandable from an evolutionary standpoint as the ultimate goal of the virus is to replicate and circulate, rather than to disappear, as observed with other pathogens [51].

Our study has some limitations, exemplified by the lack of clinical data for HAdV, EV, and NoVggII, which makes it difficult to establish a connection with confirmed infected individuals. In fact, active clinical surveillance of such viral infections is not established in Italy. Nevertheless, recent research papers have highlighted the importance of clinical surveillance for some viruses [32,52,53]. Additionally, the surveillance was conducted in a specific area of Tuscany, and the results obtained may not be generalizable to other regions. To overcome this limitation, it is recommended to expand surveillance to a larger geographic area. Furthermore, in the present paper, the presence and amount of viruses were detected using PCR as the only molecular method, but the analysis could be further improved by employing other molecular approaches that are less sensitive to environmental PCR inhibitors, such as biosensors or digital PCR [54].

## 5. Conclusions

Despite continuous monitoring of pathogens in sewage having the ability to provide early and timely data on their circulation in a population, it is expensive and time consuming. Therefore, the use of sensitive and standardized techniques, coupled with a well-organized monitoring plan, over time is worthwhile. In our study, weekly monitoring of WWTP inlets and the use of sensitive techniques allowed the identification of a seasonal trend for human adenovirus and enterovirus, with peaks in spring and summer. However, no significant temporal differences were observed for norovirus and SARS-CoV-2. For SARS-CoV-2, using the same methodologies, we have also highlighted a potential transition from epidemic to endemic status, possibly related to the development of new variants. These findings could be of great importance from a public health perspective, enabling the monitoring and prevention of potential peaks. In conclusion, WBE can be considered as an important tool to evaluate the spatio-temporal evolution of endemic–epidemic infection in the absence of clinical surveillance.

## Figures and Tables

**Figure 1 viruses-16-00482-f001:**
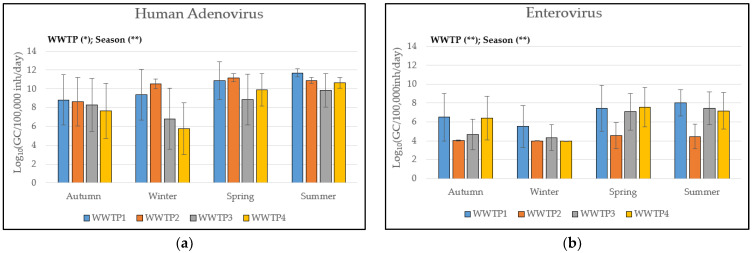
Viral load (Log_10_ GC/100,000 inh/day) observed in the WWTPs (WWTP1, WWTP2, WWTP3, WWTP4) divided according to season (autumn, spring, winter, summer) for each viral target: (**a**) HAdV, (**b**) EV, (**c**) NoVggII, and (**d**) SARS-CoV-2. Asterisk indicates statistical significance of two-way ANOVA (* = *p*-value < 0.05, ** = *p*-value < 0.01). n.s. = not significant.

**Figure 2 viruses-16-00482-f002:**
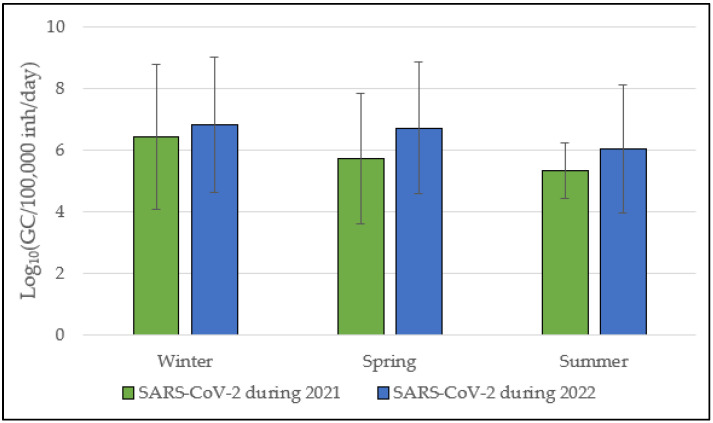
SARS-CoV-2 load during the 2021 (February 2021–September 2021) and 2022 (October 2021–September 2022) surveillance periods. Data are presented as Log_10_(GC/100,000 inh/day) mean ± standard deviation.

**Table 1 viruses-16-00482-t001:** PCR protocols for detection of HAdV, EV, NoVggII, and SARS-CoV-2.

Virus	Target Region	Primers and Probes	Sequences (5′-3′)	Reference
Human Adenovirus	Ad hexon gene	AdF	CWTACATGCACATCKCSGG	[29]
		AdR	CRCGGGCRAAYTGCACCAG	
		AdP1	FAM-CCGGGCTCAGGTACTCCGAGGCGTCCT-TAMRA	
Enterovirus	5′UTR region	EVF	GGCCCCTGAATGCGGCTAAT	[30]
		EVR	CACCGGATGGCCAATCCAA	
		EV	FAM-CGGACACCCAAAGTAGTCGGTTCCG-TAMRA	
Norovirus ggII	ORF1-ORF2 region: RNA-dependent RNA polymerase (RdRp)	JJV2F	CAAGAGTCAATGTTTAGGTGGATGAG	[31]
		COG2R	TCGACGCCATCTTCATTCACA	
		RING2-TP	FAM-TGGGAGGGCGATCGCAATCT-BHQ	
SARS-CoV-2	ORF1ab region: nsp14; 3′ to 5′ exonuclease	2297 CoV-2-F	ACATGGCTTTGAGTTGACATCT	[26,27]
		2298 CoV-2-R	AGCAGTGGAAAAGCATGTGG	
		2299 CoV-2-P	FAM-CATAGACAACAGGTGCGCTC-MGBEQ	

**Table 2 viruses-16-00482-t002:** Target viruses in the study area divided by WWTP. Positive samples are reported as number of samples in which the virus was detected out of the total observations and as percentages. The viral load is reported as Log_10_(GC/100,000 inh/day) mean ± standard deviation.

Virus		WWTP1	WWTP2	WWTP3	WWTP4	Total
Human Adenovirus	Positive samples (no., %)	42/48, 87.5%	45/48, 93.7%	36/51, 78.6%	35/50, 70%	158/197, 80.2%
	Viral load Log_10_(GC/100,000 inh/day)	10.3 ± 2.3	10.3 ± 1.6	8.5± 2.8	8.6 ± 2.9	9.4 ± 2.6
Enterovirus	Positive samples (no., %)	30/48, 62.5%	4/48, 8.3%	24/51, 47%	29/50, 58.0%	87/197, 44.2%
	Viral load Log_10_(GC/100,000 inh/day)	7.0 ± 2.3	4.3 ± 1	5.9 ± 2.1	6.6 ± 2.2	5.9 ± 2.2
Norovirus genogroup II	Positive samples (no., %)	44/48, 91.7%	33/48, 68.7%	42/51, 82.3%	47/50, 94%	166/197, 84.3%
	Viral load Log_10_(GC/100,000 inh/day)	9.1 ± 1.6	7.4 ± 2.4	8.2 ± 2.1	9.1 ± 1.4	8.5 ± 2.0
SARS-CoV-2	Positive samples (no., %)	30/48, 62.5%	21/48, 43.7%	22/51, 43.1%	27/50, 54%	100/197, 50.8%
	Viral load Log_10_(GC/100,000 inh/day)	7.2 ± 2.1	6.1 ± 1.9	6.2 ± 2.1	6.6 ± 2.1	6.5 ± 2.1

## Data Availability

Public SARS-CoV-2 wastewater dataset: https://www.iss.it/cov19-acque-reflue (accessed on 4 January 2024).

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
