# Peer review of "Wastewater-Based Epidemiology for Viral Surveillance from an Endemic Perspective: Evidence and Challenges"

_viruses, 2024, doi:10.3390/v16030482_

Round 1

Reviewer 1 Report

Comments and Suggestions for Authors

Line 18_ Real time or reverse transcriptage ?

Line 21: I do not get you “shift from an epidemic to a pandemic status, likely due to the evolution

Line 30-31: Rephrase it, opening sentence is so weak-

Line 38: correct it [2],[3],

Line 48: You need citations covering all major continents eg. https://doi.org/10.1016/j.watres.2022.118220, https://doi.org/10.1016/j.scitotenv.2020.138764, https://doi.org/10.1016/j.scitotenv.2022.159350, https://doi.org/10.1016/j.envpol.2023.122471, https://doi.org/10.1016/j.scitotenv.2020.140405, https://doi.org/10.1016/j.scitotenv.2020.140621

Table 2 and lines 206-209, Figure 1: fix the unit as GC/100 000 population/day

Line 333: Why you are so negative about continuous monitoring in conclusion, you do not think weekly monitoring is continuous monitoring?

Comments on the Quality of English Language

No major issue on English language, 

Author Response

Please consider the point-by-point response that has been attached

Reviewer 2 Report

Comments and Suggestions for Authors

The authors report the results of viral surveillance in a region. This article covers a recent topic of high importance. The reviewer suggests minor revisions before the publication.

1.     In the section 2.2, add the volume of a sample before the concentration, although the final volume (2 mL) is given.

2.     Add the definition of negative sample, because the authors conducted PCR with 45 cycles of amplification. In addition, negative samples for presence of viral genome were considered equal to the half of the limit of detection(LOD). Add values of LODs for the target viruses.

3.     In the discussion, the authors mentioned “from an epidemic to an endemic trend” in the discussion section, while they mentioned “a shift from an epidemic to a pandemic status”. The reviewer could not follow the logic for these sentences, because this study monitored virus concentrations at only four WWTPs in a region and may not provide world-wide circulation of the viruses. The authors are advised to revise the manuscript to discuss the findings and implications based on the results of this study.

Comments on the Quality of English Language

The reviewer did not find major problems in the language.

Author Response

(The authors gave the same response as above.)
